# Semi-supervised Semantic Segmentation with Prototype-based Consistency Regularization

**Hai-Ming Xu[1] , Lingqiao Liu[1]\*, Qiuchen Bian[2] , Zhen Yang[3]**
[1]Australian Institute for Machine Learning, The University of Adelaide,
[2]Northeastern University, [3] Huawei Noah's Ark Lab
{hai-ming.xu, lingqiao.liu}@adelaide.edu.au ,
bian.qiu@northeastern.edu , yang.zhen@hauwei.com

## Abstract

Semi-supervised semantic segmentation requires the model to effectively propagate the label information from limited annotated images to unlabeled ones. A challenge for such a per-pixel prediction task is the large intra-class variation, i.e., regions belonging to the same class may exhibit a very different appearance even in the same picture. This diversity will make the label propagation hard from pixels to pixels. To address this problem, we propose a novel approach to regularize the distribution of within-class features to ease label propagation difficulty. Specifically, our approach encourages the consistency between the prediction from a linear predictor and the output from a prototype-based predictor, which implicitly encourages features from the same pseudo-class to be close to at least one within-class prototype while staying far from the other between-class prototypes. By further incorporating CutMix operations and a carefully-designed prototype maintenance strategy, we create a semi-supervised semantic segmentation algorithm that demonstrates superior performance over the state-of-the-art methods from extensive experimental evaluation on both Pascal VOC and Cityscapes benchmarks[2].

## 1 Introduction

Semantic segmentation is a fundamental task in computer vision and has been widely used in many vision applications [32, 2, 29]. Despite the advances, most existing successful semantic segmentation systems [26, 6, 9, 46] are supervised, which require a large amount of annotated data, a time-consuming and costly process. Semi-supervised semantic segmentation [49, 44, 30, 20, 8, 45, 19, 38] is a promising solution to this problem, which only requires a limited number of annotated images and aims to learn from both labeled and unlabeled data to improve the segmentation performance. Recent studies in semi-supervised learning approaches suggest that pseudo-labeling [24, 1, 43] and consistency-based regularization [23, 3, 40] are two effective schemes to leverage the unlabeled data. Those two schemes are often integrated into a teacher-student learning paradigm: the teacher model generates pseudo labels to train a student model that takes a perturbed input [34]. In such a scheme, and also for most pseudo-labeling-based approaches, the key to success is how to effectively propagate labels from the limited annotated images to the unlabeled ones. A challenge for the semi-supervised semantic segmentation task is the large intra-class variation, i.e., regions belonging to the same class may exhibit a very different appearance even in the same picture. This diversity will make the label propagation hard from pixels to pixels.

In this paper, we propose a novel approach to regularize the distribution of within-class features to ease label propagation difficulty. Our method adopts two segmentation heads (a.k.a, predictors):

---

\*Corresponding author
[2]Code is available at `https://github.com/HeimingX/semi_seg_proto`.

a standard linear predictor and a prototype-based predictor. The former has learnable parameters that could be updated through back-propagation, while the latter relies on a set of prototypes that are essentially local mean vectors and are calculated through running average. Our key idea is to encourage the consistency between the prediction from a linear predictor and the output from a prototype-based predictor. Such a scheme implicitly regularizes the feature representation: features from the same class must be close to at least one class prototype while staying far from the other class prototypes. We further incorporate CutMix operation [42] to ensure such consistency is also preserved for perturbed (mixed) input images, which enhances the robustness of the feature representation. This gives rise to a new semi-supervised semantic segmentation algorithm that only involves one extra consistency loss to the state-of-the-art framework and can be readily plugged into other semi-supervised semantic segmentation methods. Despite its simplicity, it has demonstrated remarkable improvement over the baseline approach and competitive results compared to the state-of-the-art approaches, as discovered in our experimental study.

## 2 Related Work

**Semi-supervised Learning** has made great progress in recent years due to its economic learning philosophy [48]. The success of most of the semi-supervised learning researches can attribute to the following two learning schemes: pseudo-labeling and consistency regularization. Pseudo-labeling based methods [24, 5, 1, 43] propose to train the model on unlabeled samples with pseudo labels generated from the up-to-date optimized model. While consistency regularization based methods [23, 35, 37, 3, 40] build upon the *smoothness assumption* [27] and encourage the model to perform consistent on the same example with different perturbations. The recently proposed semi-supervised method FixMatch [34] successfully combine these two techniques together to produce the state-of-the-art classification performance. Our approach draws on the successful experience of general semi-supervised learning and applies it to the semi-supervised semantic segmentation task.

**Semi-supervised Semantic Segmentation** benefits from the development of general semi-supervised learning and various kinds of semi-supervised semantic segmentation algorithms have been proposed. For example, PseudoSeg method [49] utilizes the Grad-CAM [31] trick to calibrate the generated pseudo-labels for semantic segmentation network training. While CPS [8] builds two parallel networks to generate cross pseudo labels for each each. CutMix-Seg method [13] introduces the CutMix augmentation into semantic segmentation to construct consistency constraints on unlabeled samples. Alternatively, CCT [30] chooses to insert perturbations into the manifold feature representation to enforce a consistent prediction. And U$^2$PL [38] proposes to make sufficient use of unreliable pseudo supervisions. Meanwhile, considering the class-imbalance problem of semi-supervised semantic segmentation, several researches [19, 18, 14] have been published. Our approach is inspired by the observation that large intra-class variation hinders the label information propagation from pixels to pixels in semi-supervised semantic segmentation and we propose a prototype-based consistency regularization method to alleviate this problem which is novel for related literature.

**Prototype-based Learning** has been well studied in the machine learning area [16]. The nearest neighbors algorithm [11] is one of the earliest works to explore the use of prototypes. Recently, researchers have successfully used prototype-based learning to solve various problems, e.g., the prototypical networks [33] for few-shot learning and prototype-based classifier for semantic segmentation [46]. Our work further introduces the prototype-based learning into the semi-supervised problem and proves its effectiveness.

## 3 Our Approach

In this section, we first give an overview of our approach and then introduce the core concept of prototype-based consistency regularization for semi-supervised semantic segmentation. Finally, we introduce how the prototype is constructed and maintained throughout the learning process.

### 3.1 Preliminary

**Problem setting:** Given a set of labeled training images $D^l = \{(I_i^l, Y_i^l)\}_{i=1}^{N_l}$ and a set of unlabeled images $D^u = \{I_i^u\}_{i=1}^{N_u}$, where $N_u \gg N_l$, semi-supervised semantic segmentation aims to learn a

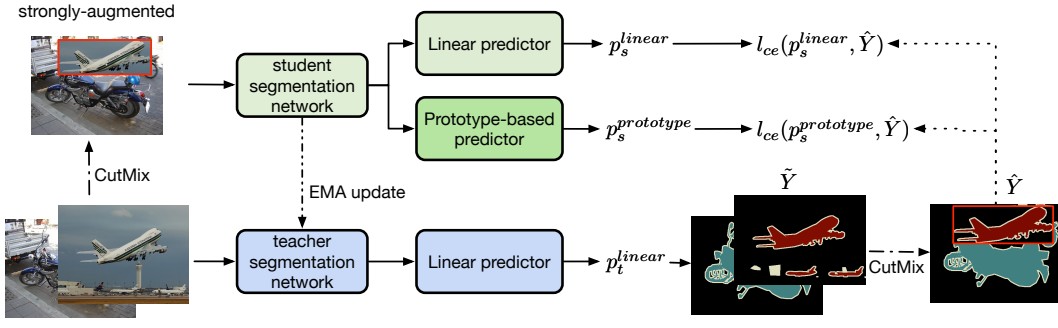

Figure 1: Overview of our method. Our method is build upon the popular student-teacher frameworks with CutMix operations. In addition to the existing modules in such a framework, we further introduce a prototype-based predictor for the student model. The output $p_s^{prototype}$ of prototype-based predictor will be supervised with the pseudo-label generated from the linear predictor of teacher model. Such kind of consistency regularization will encourage the features from the same class to be closer than the features of other classes and ease the difficulty of propagating label information from pixels to pixels. This simple modification brings a significant improvement.

segmentation model from both the labeled and unlabeled images. We use $\tilde{Y}$ denote the segmentation output and $\tilde{Y}[a, b]$ indicates the output at the $(a, b)$ coordinate.

**Overview:** the overall structure of the proposed method is shown in Figure1, our approach is built on top of the popular student-teacher framework for semi-supervised learning [35, 34, 47, 28, 43]. During the training procedure, the teacher model prediction will be selectively used as pseudo-labels for supervising the student model. In other words, the back-propagation is performed on the student model only. More specifically, the parameters of the teacher network are the exponential moving average of the student network parameters [35]. Following the common practice [34], we also adopt the weak-strong augmentation paradigm by feeding the teacher model weakly-augmented images and the student strongly-augmented images. In the context of image segmentation, we take the normal data augmentation (i.e., random crop and random horizontal flip of the input image) as the weak augmentation and CutMix [42] as the strong data augmentation.

The key difference between our method and existing methods [13, 30, 41, 8, 38] is the use of both a linear predictor (in both teacher and student models) and a prototype-based predictor (in the student model only). As will be explained in the following section, the prediction from the teacher model's linear predictor will be used to create pseudo labels to supervise the training of the prototype-based predictor of student model. This process acts as a regularization that could benefit the label information propagation.

### 3.2 Prototype-based Predictor for Semantic Segmentation

Prototype-based classifier is a long-standing technique in machine learning [21, 4]. From its early form of the nearest neighbour classifier or the nearest mean classifier to prototypical networks in the few-shot learning literature [33], its idea of using prototypes instead of a parameterized classifier has been widely adopted in many fields. Very recently, prototype-based variety has been introduced into the semantic segmentation task [46] and has been proved to be effective under a fully-supervised setting. Formally, prototype-based classifier/predictors make the prediction by comparing test samples with a set of prototypes. The prototype can be a sample feature or the average of a set of sample features of the same class. Without loss of generality, we denote the prototype set as $\mathcal{P} = \{(\mathsf{p}_i, y_i)\}$, with $\mathsf{p}_i$ indicate the prototype and $y_i$ is its associated class. Note that the number of prototypes could be larger than the number of classes. In other words, one class can have multiple prototypes for modelling its diversity. More formally, with the prototype set, the classification decision can be made by using

$$\tilde{y} = y_k \ \ s.t. \ \ k = \arg\max_i sim(x, \mathsf{p}_i), \tag{1}$$

where $sim(\cdot, \cdot)$ represents the similarity metric function, e.g., cosine distance. $\tilde{y}$ means the class assignment for the test data $x$. The posterior probability of assigning a sample to the $c$-th class can

also be estimated in prototype-based classifier via:

$$p^{prototype}(y = c|x) = \frac{\exp\left(\max_{i|y_i=c} sim(\mathbf{p}_i, x)/T\right)}{\sum_{t=1}^{C} \exp\left(\max_{j|y_j=t} sim(\mathbf{p}_j, x)/T\right)}, \quad (2)$$

where $T$ is the temperature parameter and can be empirically set. Note that Eq. 2 essentially uses the maximal similarity between a sample and prototypes of a class as the similarity between a sample and a class.

### 3.3 Consistency Between Linear Predictor and Prototype-based Predictor

Although both prototype-based classifiers and linear classifiers can be used for semantic segmentation [46], they have quite different characteristics due to the nature of their decision-making process. Specifically, linear classifiers could allocate learnable parameters[3] for each class, while prototype-based classifiers solely rely on a good feature representation such that samples from the same class will be close to at least one within-class prototypes while stay far from prototypes from other classes. Consequently, linear classifiers could leverage the learnable parameter to focus more on discriminative dimensions of a feature representation while suppressing irrelevant feature dimensions, i.e., by assigning a higher or lower weight to different dimensions. In contrast, prototype-based classifiers cannot leverage that and tend to require more discriminative feature representations.

The different characteristics of prototype-based and linear classifiers motivate us to design a loss to encourage the consistency of their predictions on unlabeled data to regularize the feature representation. Our key insight is that a good feature should support either type of classifier to make correct predictions. In addition to using two different types of classifiers, we also incorporate the CutMix [42] strategy to enhance the above consistency regularization. CutMix augmentation is a popular ingredient in many state-of-the-art semi-supervised semantic segmentation methods [8, 25, 38]. Specially, we first perform weak augmentation, e.g., random flip and crop operations, to the input images of the teacher model and obtain the pseudo-labels from the linear classifier. Next, we perform the CutMix operation by mixing two unlabeled images $mix(I_i, I_j)$ and their associated prediction $mix(\tilde{Y}_i, \tilde{Y}_j)$. The mixed image $mix(I_i, I_j)$ is fed to the student model and the output from the prototype-based classifier is then enforced to fit the pseudo-labels generated from $mix(\tilde{Y}_i, \tilde{Y}_j)$.

**Algorithm details:** As a semi-supervised segmentation algorithm, we apply different loss functions for labeled images and unlabeled images.

For a batch of labeled images $\{(I_i^l, Y_i^l)\}_{i=1}^{\mathcal{B}^l} \in D^l$, we train both the linear predictor and the prototype-based predictor. The linear classifier $\{\mathbf{w}_i\}_{i=1}^{C}$ can produce a posterior probability estimation $p_s^{linear}(Y[a, b] = c|I_i^l)$

$$p_s^{linear}(Y[a, b] = c|I_i^l) = \frac{\exp(\mathbf{w}_c^T \cdot \mathcal{F}_i^l[a, b])}{\sum_{j=1}^{C} \exp(\mathbf{w}_j^T \cdot \mathcal{F}_i^l[a, b])}, \quad (3)$$

where $\mathcal{F}_i^l[a, b] = f(A_0(I_i^l))$ means the feature extracted at location $(a, b)$ by first performing weak data augmentation $A_0$ to $I_i^l$ and then feed it to the feature extractor $f$. Meanwhile, the posterior probability of prototype-based predictor $p_s^{prototype}(Y[a, b] = c|I_i^l)$ can also be estimated via Eq. 2. We use cosine similarity for $sim(\cdot, \cdot)$ and empirically set the temperature hyperparameter $T$ to 0.1. Based on the ground truth label $Y_i^l$, the student model will be optimized by the gradient back-propagated from the two predictors simultaneously

$$\mathcal{L}_l = \mathcal{L}_l^{linear} + \mathcal{L}_l^{prototype}, \quad \text{where} \quad (4)$$

$$\mathcal{L}_l^{linear} = \frac{1}{\mathcal{B}^l} \sum_i^{\mathcal{B}^l} l_{ce}\big(p_s^{linear}(Y|I_i^l), Y_i^l\big); \quad (5)$$

$$\mathcal{L}_l^{prototype} = \frac{1}{\mathcal{B}^l} \sum_i^{\mathcal{B}^l} l_{ce}\big(p_s^{prototype}(Y|I_i^l), Y_i^l\big). \quad (6)$$

---

[3]Learnable parameters in the context means parameters that can be updated via back-propagation.

---

**Algorithm 1** Global view of our approach

---

**Inputs:**
     $D^l$: labeled set;
     $D^u$: unlabeled set;
     $T$: total number of epochs
**Outputs:**
     teacher semantic segmentation network with linear predictor only
**Process:**
 1: Prototype initialization, please refer to Algorithm 2 for details;
 2: **for** t $\leftarrow$ $[1 \rightarrow T]$ **do**
 3:    **Update student semantic segmentation network:**
 4:    Sample $B$ examples from labeled set $D^l$ and unlabeled set $D^u$, respectively;
 5:    *For labeled data*, the student model is updated based on the given ground truth, please refer to Eq.(3)-(6) of main paper;
 6:    *For unlabeled data*, weakly augmented version is fed into the teacher model to generate pseudo-labels and the student model is updated with the strongly augmented unlabeled sample based on the pseudo-labels. Please refer to Eq. (8)-(10) of main paper;
 7:    *Update prototypes* based on the ground truth of labeled samples and the pseudo-labels of unlabeled samples, please refer to Eq. (11) of main paper;
 8:    **Update teacher semantic segmentation network:**
 9:    exponential moving average (EMA) of the parameters of the student model.
10: **end for**

---

For a batch of unlabeled images $\{I_i^u\}_{i=1}^{\mathcal{B}^u} \in D^u$, we first use the teacher model to estimate their posterior probability

$$p_t^{linear}(Y[a,b] = c|I_i^u) = \frac{\exp(\mathbf{w}_c'^T \cdot \mathcal{F}_i^u[a,b])}{\sum_{j=1}^{C} \exp(\mathbf{w}_j'^T \cdot \mathcal{F}_i^u[a,b])} \tag{7}$$

where $\{\mathbf{w}_i'\}_{i=1}^C$ means the linear classifier weights of the teacher model and $\mathcal{F}_i^u[a,b] = f\big(A_0(I_i^u)\big)[a,b]$ denotes the extracted feature representation of pixel $(a,b)$ from a weakly augmented unlabeled images. Then, the class corresponding to the maximal posterior probability is the predicted class of a pixel in the given unlabeled sample, that is, $\tilde{Y}_i^u[a,b] = \arg\max_c p_t^{linear}(Y[a,b] = c|I_i^u)$. If $p_t^{linear}(\tilde{Y}[a,b]|I_i^u) \geq \tau$, where $\tau$ is a confidence threshold which is empirically set to 0.8 in our study, $\tilde{Y}[a,b]$ will be used as pseudo-labels to train the student model.

Meanwhile, for the student model we perform CutMix [42] operation among weakly augmented unlabeled samples in the same batch to create an new image (essentially, the created mix-image can be considered as a strongly-augmented image), i.e., $\hat{I}_{ij}^u = mix\big(A_0(I_i^u), A_0(I_j^u)\big)$ $s.t., \{i,j\} \in \mathcal{B}^u$, and their corresponding mixed prediction $\hat{Y}_{ij}^u = mix(\tilde{Y}_i^u, \tilde{Y}_j^u)$. Therefore, the student model can learn from the unlabeled samples through the following training objectives

$$\mathcal{L}_u = \mathcal{L}_u^{linear} + \mathcal{L}_u^{prototype}, \qquad \text{where} \tag{8}$$

$$\mathcal{L}_u^{linear} = \frac{1}{\mathcal{B}^u} \sum_{i,j \in \mathcal{B}^u} \sum_{(a,b)} l_{ce}\Big(p_s^{linear}\big(Y[a,b]|\hat{I}_{ij}^u\big), \ \hat{Y}_{ij}^u[a,b]\Big) \cdot \mathbb{1}\Big(p_t^{linear}(\hat{Y}_{ij}^u[a,b]|\hat{I}_{ij}^u) \geq \tau\Big) \tag{9}$$

$$\mathcal{L}_u^{prototype} = \frac{1}{\mathcal{B}^u} \sum_{i,j \in \mathcal{B}^u} \sum_{(a,b)} l_{ce}\Big(p_s^{prototype}\big(Y[a,b]|\hat{I}_{ij}^u\big), \ \hat{Y}_{ij}^u[a,b]\Big) \cdot \mathbb{1}\Big(p_t^{linear}(\hat{Y}_{ij}^u[a,b]|\hat{I}_{ij}^u) \geq \tau\Big) \tag{10}$$

where $p_s^{linear}(Y[a,b]|\hat{I}_{ij}^u)$ and $p_s^{prototype}(Y[a,b]|\hat{I}_{ij}^u)$ are posterior probability predictions from linear classifier and prototype-based classifier of student model respectively. Note that we use the student-teacher training for both the linear predictor and the prototype predictor, as shown in $\mathcal{L}_u^{linear}$ and $\mathcal{L}_u^{prototype}$ respectively. A global view of our approach is presented in Algorithm 1.

**Understand $\mathcal{L}_u^{prototype}$ in Eq. 10:** In order to better understand the proposed regularization loss term $\mathcal{L}_u^{prototype}$, we can consider the following significantly-simplified version of our method by omitting

---

**Algorithm 2** Prototype initialization

---

**Inputs:**
    $D^l$: labeled set
    $K$: number of prototypes per class
**Outputs:**
    initial prototypes
**Process:**
 1: **supervised training:** Train the semantic segmentation network on the subset of fully-labeled samples (please refer to Section 4.1 for training details);
 2: **feature extraction:** Use the trained segmentation network to extract feature representations of labeled samples (i.e. the feature representation before feed into the classifier of DeepLabv3+ and perform interpolation on the feature representation to match the input image size). We then sample a certain amount of pixels with their representations for each category;
 3: **feature clustering:** Perform K-Means clustering (other clustering algorithms are also possible) on sampled pixel representations from each category. This step creates $K$ sub-classes for each category. We use the feature average of samples in each subclass to obtain the initial prototypes of each category.

---

the CutMix operation: now let's imagine at a certain point of the training process, the learned feature representation can successfully support the linear classifier in making a correct prediction for some pixels. This means there are at least some discriminative feature dimensions that can distinguish classes. Without loss of generality, let's assume the feature vector for each pixel consists of two parts $\mathbf{x} = [\mathbf{x}_d, \mathbf{x}_c]$, where $\mathbf{x}_d$ is the discriminative part while $\mathbf{x}_c$ is a less discriminative part, e.g., features shared by many classes. Linear classifiers can assign lower weights to $\mathbf{x}_c$ to suppress its impact, however, the impact of $\mathbf{x}_c$ cannot be avoided by using prototype-based classifiers. Thus from the supervision of the linear classifier, the training objective of optimizing the prototype-based classifier could further suppress the generation of $\mathbf{x}_c$. Geometrically, this also encourages the features from the same class gather around a finite set of prototypes and being apart from prototypes of other classes. In this way, the (pseudo) class label can propagate more easily from pixel to pixel, which in turn benefits the learning of the linear classifier.

### 3.4 Prototype Initialization and Update

**Prototype initialization:** The prototype-based classifier does not have learnable classifier parameters but relies on a set of good prototypes. Thus it is vitally important to carefully devise strategies to initialize and maintain the pool of prototypes.

To initialize the prototypes, we first use the given labeled samples to train the semantic segmentation network (with a linear predictor) in a fully-supervised way for several epochs. Then we extract pixel-wise feature representation for each class with the trained segmentation network. With the in-class pixel-wise feature representations, we propose to perform clustering on them to find out internal sub-classes, and the initial micro-prototypes will be obtained by averaging the feature representations within the same subclass. Please find the Algorithm 2 for prototype initialization details.

**Prototype update:** In our approach, the prototypes are dynamically updated from the features extracted from the labeled images and those from unlabeled samples during the semi-supervised learning process.

When a labeled image is sampled, we assign each pixel to a prototype based on two conditions: (1) the assigned prototype $\mathsf{p}_k$ should belong to the same class as the pixel. (2) $\mathsf{p}_k$ should be the most similar prototype among all other prototypes in the same class. Once the assignment is done, we update $\mathsf{p}_k$ via

$$\mathsf{p}_k^{new} = \alpha \cdot \mathsf{p}_k^{old} + (1 - \alpha) \cdot \mathcal{F}[a, b], \tag{11}$$

where $\mathcal{F}[a, b]$ is the feature representation for the pixel at $(a, b)$. $\alpha$ is a hyper-parameter controlling the prototype update speed. We set $\alpha = 0.99$ throughout our experiment.

For unlabeled images, the ground-truth class label for each pixel is unavailable, thus we use pseudo-label instead. Recall that the pseudo-label is generated when the prediction confidence is higher than a threshold. Thus, not every pixel will be used to update the prototype.

Also, since prototype-based classifier is only used for images after the CutMix [42] operation. In our implementation, we use features extracted from the CutMix images to update the prototype rather than the original images. Empirically we find this could slightly improve the performance.

## 4 Experiments

### 4.1 Experimental Setup

Our experiment setting follows the recently proposed state-of-the-art work $U^2PL$ [38] including the evaluation datasets, semantic segmentation networks and training schedules for a fair comparison [4]. Some experimental details are listed as follows

**Datasets:** PASCAL VOC 2012 [12] is designed for visual object class recognition. It contains twenty foreground object classes and one background class. The standard partition of the dataset for training/validation/testing are 1,464/1,449/1,556 images, respectively. In the semi-supervised semantic segmentation literature, some researches [8, 19, 41, 38] also include the augmented set [15] for model training. This augmented set contains 9,118 images with coarse annotations. In the literature [38], two ways of selecting the labeled data are considered: the *classic* and the *blender* setting. The former selects labeled data from the original 1,464 candidate labeled images while the latter selects among all the 10,582 images. We evaluate our method on both settings.

Cityscapes [10] is an urban scene understanding benchmark. The initial 30 semantic classes are re-mapped into 19 classes for the semantic segmentation task. The training, validation and testing set includes 2,975, 500 and 1,525 finely annotated images respectively. For both of these two datasets, four kinds of label partitions are considered: 1/16, 1/8, 1/4 and 1/2. In this paper, we compare all methods under the identical released label splits from $U^2PL$ [38] for a fair comparison.

**Evaluation:** We use single scale cropping for the evaluation of PASCAL VOC 2012 and slide window evaluation for Cityscapes for its high resolution. The mean of Intersection over Union (mIoU) is adopted as the evaluation metric. All numbers reported in this paper are measured on the validation set of these two datasets.

**Methods:** We compare our approach with several peer-reviewed semi-supervised segmentation algorithms: Mean Teacher (NeurIPS 2017) [35], CutMix-Seg (BMVC 2020) [13], PseudoSeg (ICLR 2020) [49], CCT (CVPR 2020) [30], GCT (ECCV 2020) [20], CPS (CVPR 2021) [8], $PC^2$ Seg(ICCV 2021) [45], AEL (NeurIPS 2021) [19] and $U^2PL$ (CVPR 2022) [38]. Meanwhile, performance of supervised only on labeled data is also reported for a reference baseline. To make a fair comparison, we conduct all experiments based on the same codebase released by the authors of $U^2PL$ [38].

**Implementation Details:** Following the common practice, we use ResNet-101 [17] pre-trained on ImageNet [22] as our backbone and DeepLabv3+ [7] as the decoder. We take the default segmentation head as the pixel-level linear classifier. The feature representations for constructing the prototypes of our approach are extracted from the output of ASPP module [6]. Our experiments were run on 8 * NVIDIA Tesla V100 GPUs (memory is 32G/GPU).

For both datasets, we adopt stochastic gradient descent (SGD) as the optimizer and set batch size to 16 for model optimization. While other training details are slightly different, e.g., PASCAL VOC 2012 is trained with initial learning rate $1.0 \times 10^{-3}$, weight decay $1.0 \times 10^{-4}$ and 80 training epochs; while Cityscapes is trained with initial learning rate $1.0 \times 10^{-2}$, weight decay $5.0 \times 10^{-4}$ and 200 training epochs. Meanwhile, we use the polynomial policy to dynamically decay the learning rate along the whole training: $lr = lr_{init} \cdot (1 - \frac{iter}{totaliter})^{0.8}$.

### 4.2 Comparison with State-of-the-Arts

**Results on PASCAL VOC 2012 Dataset [12]:** Table 1 and Table 2 report the comparison results on PASCAL VOC 2012 validation set under different label quality settings. First, the results in Table 1 are obtained under the *classic* setting and our approach achieves consistent performance improvements over the compared methods. Specifically, our method outperforms the Supervised Only baseline by a large margin especially for the fewer data settings, e.g., **+24.29%** for 1/16 and

---

[4]`https://github.com/Haochen-Wang409/U2PL` (Apache 2.0 license)

Table 1: Comparing results of state-of-the-art algorithms on **PASCAL VOC 2012** `val` set with mIoU (%) ↑ metric. Methods are trained on the ***classic*** setting, i.e., the labeled images are selected from the original VOC `train` set, which consists of $1,464$ samples in total.

| Method | 1/16 (92) | 1/8 (183) | 1/4 (366) | 1/2 (732) | Full (1464) |
|---|---|---|---|---|---|
| Supervised Only | 45.77 | 54.92 | 65.88 | 71.69 | 72.50 |
| Mean Teacher [35] | 51.72 | 58.93 | 63.86 | 69.51 | 70.96 |
| CutMix-Seg [13] | 52.16 | 63.47 | 69.46 | 73.73 | 76.54 |
| PseudoSeg [49] | 57.60 | 65.50 | 69.14 | 72.41 | 73.23 |
| $PC^2Seg$ [45] | 57.00 | 66.28 | 69.78 | 73.05 | 74.15 |
| $U^2PL$ [38] | 67.98 | 69.15 | 73.66 | 76.16 | 79.49 |
| **Ours** | **70.06** | **74.71** | **77.16** | **78.49** | **80.65** |

Table 2: Comparing results of state-of-the-art algorithms on **PASCAL VOC 2012** `val` set with mIoU (%) ↑ metric. Methods are trained on the ***blender*** setting, i.e., the labeled images are selected from the augmented VOC `train` set, which consists of $10,582$ samples in total.

| Method | 1/16 (662) | 1/8 (1323) | 1/4 (2646) | 1/2 (5291) |
|---|---|---|---|---|
| Supervised Only | 67.87 | 71.55 | 75.80 | 77.13 |
| Mean Teacher [35] | 70.51 | 71.53 | 73.02 | 76.58 |
| CutMix-Seg [13] | 71.66 | 75.51 | 77.33 | 78.21 |
| CCT [30] | 71.86 | 73.68 | 76.51 | 77.40 |
| GCT [20] | 70.90 | 73.29 | 76.66 | 77.98 |
| CPS [8] | 74.48 | 76.44 | 77.68 | 78.64 |
| AEL [19] | 77.20 | 77.57 | 78.06 | 80.29 |
| $U^2PL$ [38] | 77.21 | 79.01 | 79.30 | 80.50 |
| **Ours** | **78.60** | **80.71** | **80.78** | **80.91** |

**+19.79%** for 1/8 setting respectively. Meanwhile, our approach also successfully beats other semi-supervised methods. Taking the recently proposed state-of-the-art method $U^2PL$ [38] as an example, the performance gain of our approach reaches to **+5.56%** and **+3.50%** mIoU improvements under 1/8 and 1/4 label partitions, respectively.

Table 2 presents comparison results on the *blender* setting. It is clear that our proposed method still achieves overall significant improvement over all other baselines. For example, our method excels to the Supervised Only baseline over **10%** mIoU on the 1/16 split. Compared with previous well performed algorithms, e.g., AEL [19] and $U^2PL$ [38], our approach yields superior segmentation performance, e.g., **+1.39%**, **+1.70%** and **+1.48%** on 1/16, 1/8 and 1/4 label partitions respectively.

**Results on Cityscapes Dataset [10]:** Table 3 provides comparison results of our method against several existing algorithms on Cityscapes validation set. Compared to Supervised Only baseline, our method achieves a great performance improvement due to the make use of unlabeled data, e.g., under the 1/16 label partition, our approach surpasses Supervised Only baseline by **7.67%**. Then, compared to the simple Mean Teacher [35] baseline, our approach also performs better in all cases. Furthermore, our approach is superior than the state-of-the-art algorithm $U^2PL$ [38], e.g., **Ours** excels to $U^2PL$ by **3.11%**, **1.94%** and **1.93%** under the 1/16, 1/8 and 1/4 label partition, respectively.

Note that our method performs slightly worse than AEL [19] on the 1/16 label partition, it is because the class imbalance issue is more severe on this partition, and the AEL method, which is specially designed for handling the class imbalance problem, thus gains greater improvement. Since the purpose of this paper is to explore the new consistency loss to alleviate intra-class variation for the semi-supervised semantic segmentation task, we do not explicitly consider measures to handle the label imbalance issue. Theoretically, the techniques for solving label imbalance issues can also be incorporated into our method for optimizing the overall performance.

Table 3: Comparing results of state-of-the-art algorithms on **Cityscapes** `val` set with mIoU (%) ↑ metric. Methods are trained on identical label partitions and the labeled images are selected from the Cityscapes `train` set, which consists of $2,975$ samples in total.

| Method | 1/16 (186) | 1/8 (372) | 1/4 (744) | 1/2 (1488) |
|---|---|---|---|---|
| Supervised Only | 65.74 | 72.53 | 74.43 | 77.83 |
| Mean Teacher [35] | 69.03 | 72.06 | 74.20 | 78.15 |
| CutMix-Seg [13] | 67.06 | 71.83 | 76.36 | 78.25 |
| CCT [30] | 69.32 | 74.12 | 75.99 | 78.10 |
| GCT [20] | 66.75 | 72.66 | 76.11 | 78.34 |
| CPS [8] | 69.78 | 74.31 | 74.58 | 76.81 |
| AEL [19] | **74.45** | 75.55 | 77.48 | 79.01 |
| U$^2$PL [38] | 70.30 | 74.37 | 76.47 | 79.05 |
| **Ours** | 73.41 | **76.31** | **78.40** | **79.11** |

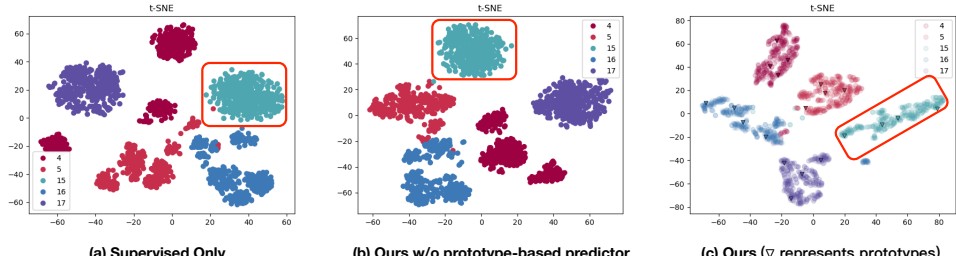

(a) Supervised Only  (b) Ours w/o prototype-based predictor  (c) Ours (∇ represents prototypes)

Figure 2: Feature embedding visualizations of (a) Supervised Only, (b) ours without prototype-based predictor and (c) our method on the 1/16 partition of Pascal VOC 2012 using t-SNE [36]. As the data distribution shown in the red boxes, within-class feature representation of our method is more compact than the ones of the Supervised Only baseline and that of the variant without prototype-based predictor, which thus alleviates the large intra-class variation problem and eases the label information propagation from pixels to pixels. The corresponding relationship between the displayed category ID and semantic category is: {4: "boat", 5: "bottle", 15: "person", 16: "pottedplant", 17: "sheep"}.

## 4.3 Ablation Study

To investigate how our approach works on the semi-supervised semantic segmentation task, we conduct ablation studies on the *classic* PASCAL VOC 2012 setting under 1/16 and 1/8 partition protocols from the following perspective views:

**Effectiveness of different components**: Table 4 presents ablation studies of several variants of our approach based on the usage of different components. The variant ④, which uses of all components, is the default setting of our approach and is presented here for a reference. The variant ① only contains a linear predictor and the prototype-based predictor is omitted. It is clear that the performance of this variant drops a lot compared to our approach and this proves that the prototype-based predictor plays a key role in our approach. On the contrary, the variant ② only maintains a prototype-based predictor and dynamically updates the prototypes during the training. The corresponding results are shown to be the worst among all the compared variants in Table 4. We postulate the potential reason is that the prototype-based predictor itself is not good enough to generate high quality pseudo-labels without the help of the linear classifier under the limited labeled data setting and thus cannot fully leverage the large amount of unlabeled samples. The variant ③ ablates the necessity of prototype update in our approach and the performance gap between this variant and variant ④ shows that our approach will benefit from the prototype update procedure and produce overall best performance.

**Distribution of feature representation**: The core idea of introducing prototype-based predictor in our approach is to utilize the prototype-based consistency regularization for alleviating the strong intra-class variation problem in semi-supervised semantic segmentation. Therefore, we are interested

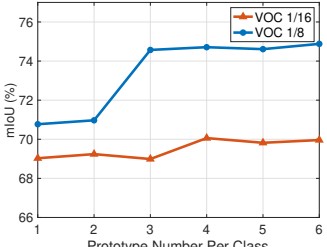

Figure 3: Abl. stu. number of prototype.

Table 4: Ablation study on the effectiveness of different components of our approach.

|   | linear pred. | proto. pred. | update proto. | 1/16 | 1/8 |
|---|---|---|---|---|---|
| ① | ✓ | | | 67.95 | 70.99 |
| ② | | ✓ | ✓ | 65.15 | 66.10 |
| ③ | ✓ | ✓ | | 67.53 | 71.89 |
| ④ | ✓ | ✓ | ✓ | **70.06** | **74.71** |

in the influence of our method on feature distribution. Figure 2 presents the feature distribution of various methods for some classes of Pascal VOC 2012. We can find that our method tends to produce more compact feature distributions than other compared methods for every semantic class and such compact feature will ease the label information propagation from pixels to pixels and thus weaken the influence of intra-class variation.

**Number of prototype**: For the prototype-based classifier, the number of prototype is not restricted to be equal to the number of classes. In our approach, we construct multiple prototypes for each semantic class to handle the intra-class variation problem of semi-supervised semantic segmentation task. In order to explore the influence of the number of prototypes on our method, we conduct ablation studies on our approach with different number of prototypes. As the results shown in Figure 3, the performance is tend to be saturate when the prototype number reaches to 4 for each semantic class. Therefore, we empirically take this number as the default value of our approach.

## 5 Limitations

One underlying assumption about our approach is that we mainly consider convolutional based semantic segmentation networks. Recently transformer-based algorithms [9, 39] are being investigated for semantic segmentation that are not explored in this paper and is left for future work. One underlying assumption about our approach is that we mainly consider semantic segmentation networks of per-pixel prediction style.

## 6 Conclusion

Semi-supervised semantic segmentation aims to propagate label information from pixels to pixels effectively, but the large intra-class variation hinders the propagation ability. In this paper, we introduce a prototype-based predictor into our semi-supervised semantic segmentation network and propose a novel prototype-based consistency loss to regularize the intra-class feature representation to be more compact. Experimental results show that our method successfully achieves superior performance than other approaches.

## 7 Impacts and Ethics

This paper proposes a method for semi-supervised semantic segmentation which is a fundamental research topic in computer vision area and no potential negative societal impacts are known up to now. In terms of ethics, we do not see immediate concerns for the models we introduce and to the best of our knowledge no datasets were used that have known ethical issues.

**Acknowledgements** This work is partially supported by Centre of Augmented Reasoning of the University of Adelaide. Meanwhile, we would like to thank the anonymous reviewers for their insightful comments. We also gratefully acknowledge the support of MindSpore[5], CANN (Computer Architecture for Neural Networks) and Ascend AI Processor used for this research.

---

[5]https://www.mindspore.cn/

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
