# Appendix of "Semi-supervised Semantic Segmentation with Prototype-based Consistency Regularization"

**Hai-Ming Xu[1] , Lingqiao Liu[1]\*, Qiuchen Bian[2] , Zhen Yang[3]**
[1]Australian Institute for Machine Learning, The University of Adelaide,
[2]Northeastern University, [3] Huawei Noah's Ark Lab
`{hai-ming.xu, lingqiao.liu}@adelaide.edu.au` ,
`bian.qiu@northeastern.edu` , `yang.zhen@hauwei.com`

In this appendix, we first present quantitative metrics for comparing the intra-/inter-class discrimination of various methods. Next, we provide another two ablation studies to further inspect our approach. Finally, we further visualize the semantic segmentation results of our approach for better understanding.

## A Comparing of Intra-/Inter-class Discrimination

In the main paper, the visualization of feature distribution in Figure 2 (c) has demonstrated that our approach can encourage a more compact within-class feature distribution and thus ease the large intra-class variation problem in the semi-supervised semantic segmentation. In order to have quantitative comparison, we borrow the principle of linear discriminant analysis (LDA) and calculate the intra-/inter-class variance of the feature representations for each comparing methods. As the results shown in Table 1, our approach has not only improved the intra-class variance but also the inter-class variance, and thus the overall discrimination.

Table 1: Comparison of intra-/inter-class discrimination. `var.` means the variance matrix.

| *classic* VOC 2012(1/16 setting) | $\frac{tr(\text{inter-class var.})}{tr(\text{intra-class var.})} \uparrow$ | $tr(\text{inter-class var.}) \uparrow$ | $tr(\text{intra-class var.}) \downarrow$ |
|---|---|---|---|
| $U^2$PL | 0.48 | 80.78 | 168.30 |
| Ours w/o prototype-based classifier | 0.45 | 76.01 | 168.92 |
| Ours | 2.22 | 283.43 | 127.63 |

## B Ablation Studies

### B.1 Strong Data Augmentation

In the main paper, our approach is built upon the popular student-teacher weak-strong augmentation framework and the CutMix [3] strong data augmentation is utilized as the default setting. In order to further investigate the effectiveness of our approach, we conduct an ablation study by varying the data augmentation approaches while keeping other modules unchanged in any comparing methods. As results shown in Table 2, our method can still achieve overall best segmentation results with different strong data augmentations.

---

\*Corresponding author

36th Conference on Neural Information Processing Systems (NeurIPS 2022).

Table 2: Ablation study to the strong data augmentation on *classic* PASCAL VOC 2012 1/16 setting.

| strong data augmentation | Cutout [1] | ClassMix [2] |
|---|---|---|
| U$^2$PL | 66.82 | 67.77 |
| Ours w/o prototype-based classifier | 66.86 | 66.93 |
| Ours | 69.24 | 69.36 |

## B.2 Confidence Threshold

We are also interested in how our approach will be performed when various confidence thresholds are selected. From the result shown in Table 3, we find that our approach can achieve good performance when the confidence threshold falls into a reasonable range, e.g., [0.75, 0.95].

Table 3: Ablation study of sensitivity of our approach to the selection of confidence threshold on *classic* PASCAL VOC 2012 1/16 setting.

| confidence threshold | 0.95 | 0.90 | 0.85 | 0.80 | 0.75 | 0.70 |
|---|---|---|---|---|---|---|
| Linear classifier | 71.01 | 70.97 | 70.30 | 70.06 | 69.43 | 64.89 |
| Prototype-based classifier | 70.72 | 70.74 | 70.10 | 69.89 | 68.92 | 64.68 |

## C   Semantic Segmentation Visualization

In our main paper, we have verified the effectiveness of our proposed method through extensive quantitative comparative experiments. In the appendix, we want to provide more qualitative results to further support our conclusion.

Figure 1 and Figure 2 present the segmentation results of comparing methods on the PASCAL VOC 2012 validation set from the perspective of object boundary perception and object intra-class prediction consistency, respectively. Specifically, Figure 1 illustrates that our method can produce better segments for the boundary of objects. As the highlighted region shown in yellow dotted boxes, i.e., the lower edge of train (row 1), the body of person (row 2-3), the wing of airplane (row 4) and the bottle (row 5), the generated segments are much more precise for our method than the baseline method without prototype-based consistency regularization constraints.

Similarly, Figure 2 demonstrates that our method can achieve consistent category prediction within the objects, while the comparison method may always predict different parts of the same object into different categories (e.g., the dog at row 1, the train at row 2, the cow at row 3 and the cat at row 4) and sometimes even completely wrong prediction for the whole object (the sofa at row 5 and the cow at last row are completely mispredicted as chair and horse, respectively).

The superior semantic segmentation performance of our approach is attributed to the proposed prototype-based consistency regularization which encourages the features from the same class to be close to at least one within-class prototype while staying far away from the other between-class prototypes. Such kind of constraints will ease the label information propagation from pixels to pixels for the semi-supervised semantic segmentation task and therefore our approach can produce more precise segments and predict consistent categories within the same segment.

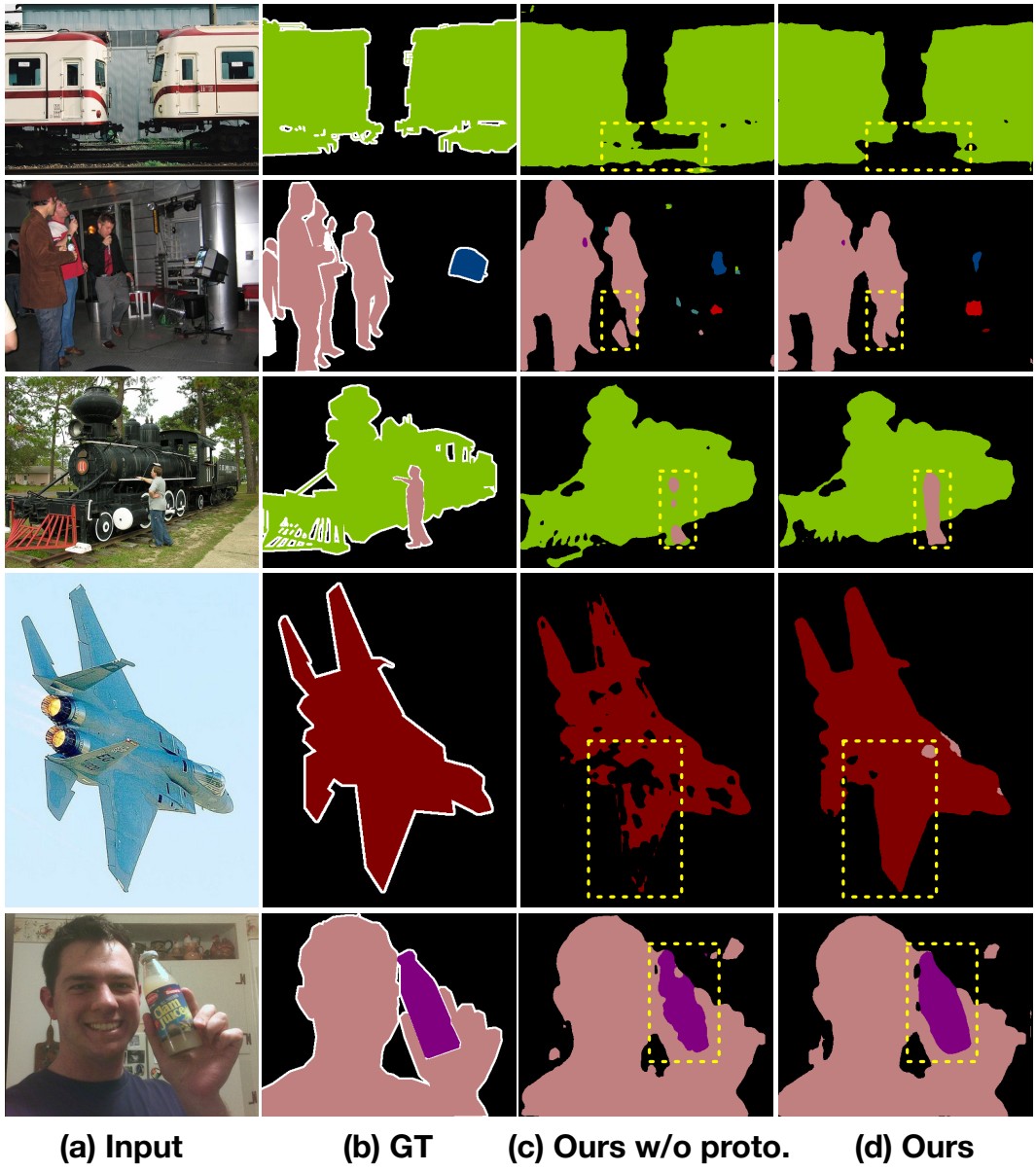

**(a) Input**    **(b) GT**    **(c) Ours w/o proto.**    **(d) Ours**

Figure 1: Qualitative results on PASCAL VOC 2012 validation set. Methods are trained on the 1/16 label partition protocol of the *classic* setting. (a) Input image, (b) Ground-truth, (c) Ours without prototype-based predictor and (d) our method. Yellow dotted boxes highlight the segments where our method performs better than the comparison method, i.e., our method can better perceive the boundary of objects.

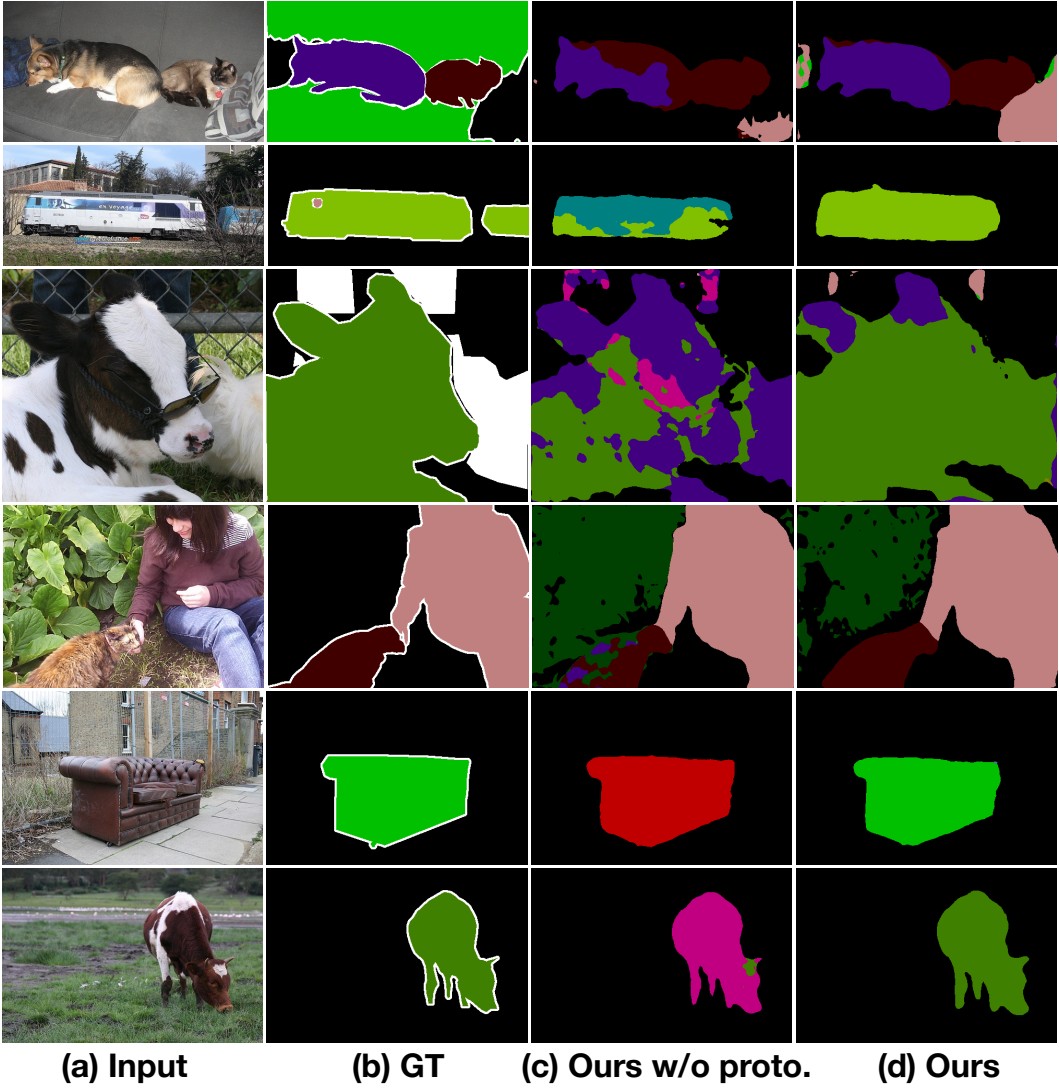

**(a) Input**  **(b) GT**  **(c) Ours w/o proto.**  **(d) Ours**

Figure 2: Qualitative results on PASCAL VOC 2012 validation set and all methods are trained on the 1/16 label partition protocol of the *classic* setting. Although both comparison methods can roughly segment the outline of the object, our approach can achieve better consistency of category prediction inner the object, especially for the objects whose appearance vary a lot, e.g., the dog at row 1 and the bus at row 2.