# OpenReview forum: "Semi-supervised Semantic Segmentation with Prototype-based Consistency Regularization"
_NeurIPS.cc/2022/Conference — NeurIPS 2022 Accept_

### Official Review · Reviewer_sBQB · 2022-06-29

**Rating:** 7
**Confidence:** 4
**Soundness:** 3 good
**Presentation:** 3 good
**Contribution:** 3 good

**Summary:**

This paper proposes a consistency based scheme for semi-supervised semantic segmentation, where they employ a student-teacher framework and enforce consistency between predictions made by a prototypical classifier and a linear classifier. By selecting multiple prototypes for each class, it is hypothesized that the limitation of intra-class distances can be addressed.

**Questions:**

- Overall, while I appreciate the simplicity and effectiveness of the method, it is not clear why this method has to work. What part of your pipeline is the most effective and makes the most difference compared to prior work? This analysis is not present in the paper. I am willing to raise my rating if this is cleared.

- What is done at test-time? If I understand correctly, you use the student model along with the classifier based predictions at test-time? How does the performance vary if we use the prototype based prediction instead?

- How is the teacher model initialized and updated?

- From fig2, any reason why 3-4 prototypes should work better instead of just one? Wouldn't the intra-class separation be least when all features are clustered around one center rather than distributed around 3-4 centers?

- Are the prototypes computed in the feature space? If so, I believe that the feature space for segmentation models is of lesser resolution - how do you get pixel level labels at this reduced resolution to compute class wise prototypes?

- It would also be interesting to visualize the prototypes themselves along with the features in the same space, and know what is actually being learnt by the prototypes and the classifier. I am more curious to know how they are different. Assume a case when all prototypes belonging to a class match the classifier weight vector for that class. All the conditions would be satisfied, but the consistency criterion would be trivial in that case.

**Limitations:**

The societal impact has been addressed. The limitations section could be better. For example, the benefits from your method seem limited if more unlabeled data is available. Do you have an intuition why?

**Strengths And Weaknesses:**

-- Strengths

- The idea of using complementary training signals from linear classifier and prototypical classifier to enforce consistency is interesting.
- Apart from a few required clarifications (see below), the paper is well written and easy to follow, and the intuitions and concepts have been explained well.
- The performance on reported datasets is extremely strong. I am indeed surprised that a method as simple as this results in such strong performance and large improvements.

-- Weaknesses

- While the abstract and the intuition is built upon the fact that this method reduces "intra-class" discrepancies, it has not been explained or demonstrated that this is indeed a limitation in prior methods.  It has also not been shown that this method better alleviates this issue compared to prior methods. For example, is there a concrete metric that helps us understand that existing methods really perform poorly on intra-class differences, while your method does better? The tSNE plots are inconclusive (the highlighted cluster already looks compact in first two plots as well, and the cluster for label "4" still forms two distinct clusters using your method, same as "supervised-only").

- There are really two parts to this framework - one is FixMatch style teacher-student model with weak-strong data augmentation, while another is the proposed consistency loss. Each of these have to evaluated separately to better understand the benefits. Currently, ablations are only present on the consistency part.

-- Minor
- Authors are encouraged to use the published versions in citations instead of the arxiv-only versions in an updated version.

---

> ### Author Response · Authors · 2022-08-02
> **Quantitative Metric and Ablation Studies (part 1)**
>
> Thank you for your positive review and useful remarks! Below please find our responses
>
> > Weakness1: [...] is there a concrete metric that helps us understand that existing methods really perform poorly on intra-class differences, while your method does better? [...]
>
> Thank you for your suggestions and we can borrow the principle of linear discriminant analysis(LDA) and calculate the intra-/inter- class variance of the feature representations for each comparing method. The results shown in the following table reveal that our method has not only improved the intra-class variance but also the inter-class variance, and thus the overall discrimination.
>
> | Classic VOC(1/16 setting)  | $\frac{tr(\text{inter-class var})}{tr(intra-class var)} \uparrow$  | $tr(\text{inter-class var})\uparrow$ | $tr(\text{intra-class var})\downarrow$ |
> | :---          | :----: | :----: | :----: |
> | U2PL | 0.48 | 80.78 | 168.30 |
> | Ours w/o prototype-based classifier |0.45 |76.01| 168.92 |
> | Ours | 2.22 | 283.43 | 127.63 |
>
>
> > Weakness2: There are really two parts to this framework - one is FixMatch style teacher-student model with weak-strong data augmentation, while another is the proposed consistency loss. Each of these have to evaluated separately to better understand the benefits. Currently, ablations are only present on the consistency part.
>
> Our method is based upon the student-teacher weak-strong augmentation framework and our main contribution is the new consistency regularization loss. That’s why we focus more on the consistency part. To evaluate the impact of the first part, we conducted an experiment by varying the data augmentation approaches while keeping the consistency part unchanged. The results shown in the following table present that our method can still achieve overall best segmentation results with different strong data augmentations.
>
> | Classic VOC (1/16 setting)         | Cutout[1] | ClassMix[2]|
> | :---                                  | :----: | ---: |
> | U2PL                                  | 66.82 | 67.77 |
> | Ours w/o prototype-based classifier   | 66.86 | 66.93 |
> | Ours                                  | 69.24 | 69.36 |
>
> [1] Terrance DeVries and Graham W Taylor. ''Improved regularization of convolutional neural networks with cutout.'' CoRR, abs/1708.04552, 2017.
>
> [2] Viktor Olsson, Wilhelm Tranheden, Juliano Pinto, Lennart Svensson. ''ClassMix: Segmentation-Based Data Augmentation for Semi-Supervised Learning'', WACV2021
>
> > Weakness3: Authors are encouraged to use the published versions in citations instead of the arxiv-only versions in an updated version.
>
> Thanks for your suggestions and we have updated the references in our paper.
>
> > Question1: [...] why this method has to work.  What part of your pipeline is the most effective and makes the most difference compared to prior work? [...]
>
> The newly proposed prototype-based consistency regularization is the main contribution in our paper and it makes the most difference compared to prior works (Line96-101 of our paper presents the difference between our proposed approach and existing methods). The performance presented in Table.1, 2 and 3 of our paper already demonstrate the superiority of our work compared to previous state-of-the-art methods. The ablation studies in Table 4 of our paper further verify the effectiveness of the proposed prototype-based consistency regularization module in our approach (comparing ① and ④, the prototype-based consistency regularization module brings 2.1% and 3.7% performance improvement on the classic PASCAL VOC 2012 1/16 and 1/8 setting, respectively).
>
> > Question2: What is done at test-time? If I understand correctly, you use the student model along with the classifier based predictions at test-time? How does the performance vary if we use the prototype based prediction instead?
>
> We use the teacher model with the linear classifier based predictions at test-time and the prototype classifier will be discarded after training. This is consistent with most of the comparing methods, e.g., U2PL[3], AEL[4] and CPS[5], which also use the teacher model for prediction at test-time and thus the comparison is fair. The prototype-based performance has been presented in the following table and It is obvious that the prototype-based classifier can also achieve comparable performance (but slightly worse) to the linear based classifier.
>
> | Classic VOC  | 1/16(92)  | 1/8(183) | 1/4(366) | 1/2(732) | full(1464) |
> | :---          | :----: | :----: | :----: |:----: |:----: |
> | Linear classifier | 70.06 | 74.71 | 77.16 | 78.49 | 80.65
> | Prototype-based classifier | 69.89| 74.31| 77.01 |77.94 | 79.82
>
> [3] Yuchao Wang, et al. ‘‘Semi-supervised semantic segmentation using unreliable pseudo-labels’’. CVPR2022.
>
> [4] Hanzhe Hu, et al. ‘‘Semi-supervised semantic segmentation via adaptive equalization learning’’. NeurIPS 2021.
>
> [5] Xiaokang Chen, et al. ''Semi-supervised semantic segmentation with cross pseudo supervision''. CVPR2021.

---

> ### Author Response · Authors · 2022-08-02
> **Quantitative Metric and Ablation Studies (part 2)**
>
> > Question3: How is the teacher model initialized and updated?
>
> The initialization and updating of the teacher model in our work is identical to the ways used in the popular semi-supervised learning method Mean-Teacher approach[6]. Specifically, at the beginning of model training, parameters of the teacher model are initialized with parameters of the student model. During the model training, parameters of the teacher model are the exponential moving average of the parameters of the up-to-date student model.
>
> [6] Antti Tarvainen, Harri Valpola. ''Mean teachers are better role models: Weight-averaged consistency targets improve semi-supervised deep learning results''. NeurIPS2017
>
> > Question4: From fig2, any reason why 3-4 prototypes should work better instead of just one? Wouldn't the intra-class separation be least when all features are clustered around one center rather than distributed around 3-4 centers?
>
> As the description at Line28-31 of our paper, per-pixel dense prediction task, e.g., the semi-supervised semantic segmentation task, may suffer from the large intra-class variation problem. We choose to use multiple prototypes per class to capture this intra-class variation rather than forcefully eliminate this variation by only using a single prototype. In the literature, it has been observed that eliminating intra-class variation at the training stage may lead to a poorly generalized prediction model, which is known as neural collapse[7].
>
> [7] Papyan, V., Han, X. Y., and Donoho, D. L. ''Prevalence of neural collapse during the terminal phase of deep learning training''. Proceedings of the National Academy of Sciences, 117(40):24652–24663, 2020.
>
> > Question5: Are the prototypes computed in the feature space? If so, I believe that the feature space for segmentation models is of lesser resolution - how do you get pixel level labels at this reduced resolution to compute class wise prototypes?
>
> Yes and Line239-240 presents how prototypes are generated from, i.e., the prototypes are computed based on the feature representations before feeding into the classifier module of DeepLabv3+. The resolution of feature representation is a quarter of the resolution of the input image. In order to assign pixel-level labels to the feature representations, we conduct an interpolation on the feature representation to transform to the input image size.
>
> > Question6: It would also be interesting to visualize the prototypes themselves along with the features in the same space, and know what is actually being learnt by the prototypes and the classifier. I am more curious to know how they are different. Assume a case when all prototypes belonging to a class match the classifier weight vector for that class. All the conditions would be satisfied, but the consistency criterion would be trivial in that case.
>
> Thanks for your suggestions and we have updated Figure2 ( c ) of our manuscript. As the Figure2 ( c ) shown, the learned micro-prototypes spread among the distribution of each semantic class which demonstrates that these micro-prototypes have already captured diversified representations for the same semantic class due to the large intra-class variation caused by the appearance changing.
> Regarding the trivial case proposed by the Reviewer sBQB, the consistency criterion will never become trivial under the student-teacher weak-strong data augmentation framework. In such a framework, a strongly augmented input (not the one used for generating the pseudo-label) will be fed into the student network and thus the output logits cannot be guaranteed to be identical to the teacher model. The same situation is true for our proposed consistency loss.
>
> > Limitations: [...] the benefits from your method seem limited if more unlabeled data is available. Do you have an intuition why?
>
> In our paper, we mainly report experimental results of varying numbers of labeled samples (e.g., 1/16, 1/8, 1/4 and 1/2 of the training set are selected to construct the labeled set, respectively) but not the unlabeled ones. According to the results, performance improvement of our approach with more labeled samples is not as big as with fewer labeled data. We think the potential reason is that more finely annotated labeled samples will alleviate the challenge of large intra-class variation problem and ease the label-propagation from labeled pixels to unlabeled ones, and thus the benefits of the proposed approach become less prominent.

---

> > ### Comment · Reviewer_sBQB · 2022-08-05
> > **Response to the rebuttal**
> >
> > I thank the authors for responding to each of my queries. Most of my questions have been addressed, and I am updating my rating based on the responses.

---

> > > ### Author Response · Authors · 2022-08-09
> > > **Thanks for Prompt Response**
> > >
> > > We sincerely thank Reviewer sBQB for your prompt response.

---

> > > > ### Comment · Reviewer_sBQB · 2022-08-09
> > > > **Raising the score.**
> > > >
> > > > I have read the author responses to questions raised by other reviewers, and decided to further increase the rating. I am highly impressed by the simplicity of this approach (à la x-Match style works in SSL), and very surprised that this simple method can achieve the results it does. This line of thought deserves deeper scrutiny from the community, and hence I think this should be accepted.

---

### Official Review · Reviewer_P8JB · 2022-07-10

**Rating:** 5
**Confidence:** 5
**Soundness:** 3 good
**Presentation:** 3 good
**Contribution:** 3 good

**Summary:**

The paper proposes a novel approach to regularize the distribution of intra-class features to ease label propagation difficulty. In particular, the proposed method adopts a standard linear predictor and a prototype-based predictor and encourages the consistency between predictions from two predictors. Experimental results validate the effectiveness of the proposed method.

**Questions:**

Please see weakness above.

**Limitations:**

Yes, the authors have addressed the limitations and potential negative social impact.

**Strengths And Weaknesses:**

Strength:

The paper is easy to follow. The proposed method is effective. Experimental results are promising.

Weakness:

(1) Limited novelty. From my point of view, the paper only proposes a consistency regularization method on the prediction head. The contribution is quite limited.

(2) Prototype generation. How to deal with false prediction on unlabeled images?

(3) Experimental results. The results of previous methods on Cityscapes dataset seem strange to me. They are different from the original results of the papers.

(4) Feature visualization. Although intra-class compactness is improved, inter-class discrimination is weakened. Will this have bad influence on segmentation?

---

> ### Author Response · Authors · 2022-08-02
> **Novelty and Clarification**
>
> Thank you for your review and useful remarks! Below please find our responses
>
> > Weakness1: Limited novelty. From my point of view, the paper only proposes a consistency regularization method on the prediction head. The contribution is quite limited.
>
> Comparing with the baseline model, our method only introduces a simple auxiliary prototype predictor (will be removed after training) and an additional consistency regularization loss. With them, we show that we can achieve significant performance improvement. We believe that this is an interesting discovery and could be valuable for advancing semi-supervised segmentation research. Moreover, we want to highlight that the contribution of many recent semi-supervised learning methods is “only proposes a consistency regularization”, e.g., see examples in the following papers. In our view, a simple-but-effective consistency regularization should be considered as a merit rather than a disadvantage.
>
> [1]Temporal ensembling for semi-supervised learning, ICLR2017
>
> [2] Virtual Adversarial Training:a Regularization Method for Supervised and Semi-supervised Learning, TPAMI2017
>
> [3]Mean teachers are better role models: Weight-averaged **consistency** targets improve semi-supervised deep learning results. NeurIPS2017
>
> [4] Interpolation **Consistency** Training for Semi-Supervised Learning, IJCAI2019
>
> [5] WCP: Worst-Case Perturbations for Semi-Supervised Deep Learning, CVPR2020
>
> [6] FeatMatch: Feature-Based Augmentation for Semi-Supervised Learning, ECCV2020
>
> [7] Unsupervised Data Augmentation for **Consistency** Training, NeurIPS2020
>
> [8] FixMatch- Simplifying Semi-Supervised Learning with **Consistency** and Confidence, NeurIPS2020
>
> [9] Time-**Consistent** Self-Supervision for Semi-Supervised Learning, ICML2020
>
> [10] Adaptive **Consistency** Regularization for Semi-Supervised Transfer Learning, CVPR2021
>
> > Weakness2: Prototype generation. How to deal with false prediction on unlabeled images?
>
> Prototypes in our approach are initialized by clustering pixel representations from labeled samples (see Line 237-247) and they will be dynamically updated based on pseudo-labels of unlabeled samples (see Line 194-208). Same as most of the state-of-the-art semi-supervised learning/segmentation approaches, pseudo-labels are inevitably noisy. But the existing teacher-student semi-supervised learning framework has shown being robust towards noisy pseudo-labels. Moreover, our prototype is calculated by averaging features from a cluster. This average operation could also resist the distraction from wrongly-labeled features.
>
> > Weakness3: Experimental results. The results of previous methods on Cityscapes dataset seem strange to me. They are different from the original results of the papers.
>
> Since U2PL[11] is a recently proposed state-of-the-art semi-supervised semantic segmentation method, we directly use the same codebase of U2PL in our paper for a fair comparison.
>
> To make a fair comparison, the original U2PL paper reimplemented existing methods with an unified setting. We directly quote the evaluation results reported in U2PL for our comparing methods. In this way, we can ensure all the methods are under the exact same setting. Note that due to implementation discrepancy, e.g., the authors of AEL method[12] use random label splits in their experiments, while the authors of U2PL use a fixed labeled set for the convenience of comparison, the results reported in the original paper and in U2PL (and thus our paper) could be different.
>
> [11] Yuchao Wang, et al. ''Semi-supervised semantic segmentation using unreliable pseudo-labels''. CVPR2022
>
> [12] Hanzhe Hu, et al. ''Semi-supervised semantic segmentation via adaptive equalization learning''. NeurIPS 2021.
>
> > Weakness4: Feature visualization. Although intra-class compactness is improved, inter-class discrimination is weakened. Will this have bad influence on segmentation?
>
> Actually, the inter-class discrimination of our approach has also been improved comparing with other methods. In order to quantitatively measure the intra-/inter- class discrimination, we can borrow the principle of linear discriminant analysis(LDA) and calculate the intra-/inter- class variance of the feature representations for each comparing methods. As seen, our approach produces higher inter-class variance and thus be more discriminative.
>
> | Classic VOC(1/16 setting)  | $\frac{tr(\text{inter-class var})}{tr(intra-class var)} \uparrow$  | $tr(\text{inter-class var})\uparrow$ | $tr(\text{intra-class var})\downarrow$ |
> | :---          | :----: | :----: | :----: |
> | U2PL | 0.48 | 80.78 | 168.30 |
> | Ours w/o prototype-based classifier |0.45 |76.01| 168.92 |
> | Ours | 2.22 | 283.43 | 127.63 |

---

> ### Author Response · Authors · 2022-08-09
> **Sincerely Look Forward to Your Feedback**
>
> Dear Reviewer P8JB,
>
> Thanks again for your insightful suggestions and comments. As the deadline for reviewer-author discussion is approaching, we are glad to provide any additional clarifications that you may need.
>
> We have carefully studied your comments and added additional clarifications and analysises in our previous responses to address your concerns. We genuinely hope you could kindly check our response.
>
> We hope that our previous responses have convinced you the merits of our work. Please do not hesitate to contact us if there are other clarifications or experiments we can offer.
>
> Thank you for your time again.
>
> Best wishes,
>
> Authors

---

### Official Review · Reviewer_LwA4 · 2022-07-11

**Rating:** 5
**Confidence:** 4
**Soundness:** 3 good
**Presentation:** 3 good
**Contribution:** 3 good

**Summary:**

This paper is about semi-supervised semantic image segmentation. A moment distillation method is used and the authors proposed to use pixels only with confidence larger than 0.8 as supervision.
Experiments are done on Pascal VOC12 and Cityscapes.


**Questions:**

1. I thinks an Algorithm describing the global process is necessary. Otherwise it is difficult to reproduce the experiment results.

2. Please explain “EMA update” in Figure 1.

3. Is Prototype similar to assign more than one template in classifier layers? For example, if there are 21 category, the classifier layer is set to  512x48.

**Limitations:**

Yes

**Strengths And Weaknesses:**

[Strengths]

1. The experimental results seems pretty good from Table 1 and Table 3.
2. The combination of cutmix and distillation is interesting.


[Weakness]

1. Ablation study of the choice of confidence threshold is absent.
2. Some details are missing. For example, L189 'in a fully-supervised way for several epochs', what the exactly hyper parameters are used?
3. Eevey

---

> ### Author Response · Authors · 2022-08-02
> **Ablation study and Clarification**
>
> Thank you for your positive review and useful remarks! Below please find our responses
>
> > Weakness1: Ablation study of the choice of confidence threshold is absent.
>
> Thanks for your suggestion. We have added such an ablation study to the newly updated supplementary material. From the result, we find that our approach can achieve good performance when the confidence threshold falls into a reasonable range, e.g., [0.75, 0.95].
>
> | Classic VOC(1/16 setting)  | 0.95  | 0.90 | 0.85 | 0.80 | 0.75 | 0.70 |
> | :---          | :----: | :----: | :----: | :----: | :----: |:----: |
> | Linear classifier     | 71.01 | 70.97 | 70.30 | 70.06 | 69.43 |64.89
> | Prototype-based classifier|70.72|70.74| 70.10 | 69.89 | 68.92 |64.68
>
> > Weakness2: Some details are missing. For example, L189 'in a fully-supervised way for several epochs', what the exactly hyper parameters are used?
>
> Sorry for the confusion. L189 presents how prototypes are initialized in our approach. We train the segmentation network on given labeled samples with the same training protocols as the Supervised Only baseline, i.e., train DeepLabv3+ with `batchsize=16`, `initial learning rate=1.0*10^-3`, `weight decay=1.0*10^-4` and `80 training epochs`. The corresponding training details have been presented in Line237-247.
>
> > Questions1: I thinks an Algorithm describing the global process is necessary. Otherwise it is difficult to reproduce the experiment results.
>
> Thanks for your suggestions. We have added the following algorithm table to the supplementary material due to the space constraint of the main paper.
>
> ```
> Algorithm procedure of our approach
> Input: labeled images $D^l$, unlabeled images D^u
> Output: teacher semantic segmentation network with linear predictor only
> Process:
> 1.  Prototype initialization, please see Section 3.4 for details
> 2.  For step in range(epoch):
> 3.  	Student semantic segmentation network update:
> 4.  		Sample a batch of labeled samples and unlabeled samples;
> 5.  		For labeled data, the student model is updated based on the given ground truth, please refer to Eq.(3)-(6) of main paper;
> 6.  		For unlabeled data, weakly augmented version is fed into the teacher model to generate pseudo-labels and the student model is updated with the strongly augmented unlabeled sample based on the pseudo-labels. Please refer to Eq. (8)-(10) of main paper;
> 7.  		Update prototypes based on the ground truth of labeled samples and the pseudo-labels of unlabeled samples, please refer to Eq. (11) of main paper;
> 8.  	Teacher semantic segmentation network update: exponential moving average (EMA) of the parameters of the student model.
> ```
>
> > Question2: Please explain “EMA update” in Figure 1.
>
> The “EMA update” operation in our paper follows the approach proposed in Mean Teacher method[1] which is a popular semi-supervised algorithm, i.e., the parameters of the teacher network are updated through the exponential moving average of the parameters of the student network at each optimization step.
>
> [1] Antti Tarvainen and Harri Valpola. ''Mean teachers are better role models: Weight-averaged con- sistency targets improve semi-supervised deep learning results''. NeurIPS2017.
>
> > Question3: Is Prototype similar to assign more than one template in classifier layers? For example, if there are 21 category, the classifier layer is set to 512x48.
>
> We have conducted the experiment suggested by the Reviewer LwA4 and the results are shown in the following table. As seen, simply increasing the number of classifiers per class does not bring any improvement over the baseline. This suggests that using prototype-based classifier and our cross-predictor consistency regularization loss is the key to success rather than using multiple classifiers.
>
> | Classic VOC  | 1/16(92)  | 1/8(183) | 1/4(366) |
> | :---          | :----: | :----: | :----: |
> | single classifier layer per class | 67.95 | 70.99 | 75.43 |
> | 4 classifier layer per class |67.76|70.89| 75.41 |

---

> ### Author Response · Authors · 2022-08-09
> **Sincerely Look Forward to Your Feedback**
>
> Dear Reviewer LwA4,
>
> Thanks again for your insightful suggestions and comments. As the deadline for reviewer-author discussion is approaching. We are glad to provide any additional clarifications that you may need.
>
> We have carefully studied your comments and added additional clarifications and experiments in our previous responses to address your concerns. We genuinely hope you could kindly check our response.
>
> We hope that our previous responses have convinced you the merits of our work. Please do not hesitate to contact us if there are other clarifications or experiments we can offer.
>
> Thank you for your time again.
>
> Best wishes,
>
> Authors

---

### Official Review · Reviewer_sZ7j · 2022-07-14

**Rating:** 6
**Confidence:** 5
**Soundness:** 3 good
**Presentation:** 3 good
**Contribution:** 3 good

**Summary:**

This work focuses on the semi-supervised semantic segmentation. To solve the large intra-class variation problem, this work attempts to regularize the distribution of within-class features. The proposed methods is mainly based on the student-teacher framework with a well-designed prototype-based predictor and a widely-used linear predictor. It encourages the consistency between the linear predictor and prototype-based predictor to regularize the distribution of within-class features. Extensive experiments have been conducted to validate the effectiveness of this method. The proposed method achieves promising performance on different datasets and settings.

**Questions:**

1. Please demonstrate more details about how the regularization between prototype-based predictor and linear predictor works.
2. Please provide more details about the prototype initialization and updating.
3. Please conduct ablative experiments about the CutMix.

**Limitations:**

This work discussed the limitation on the model architectures.
Maybe it is necessary to consider the potential data distribution unbalance problem.

**Strengths And Weaknesses:**

Strengths:
1. The idea is simple and clear.
2. Comprehensive experiments have conducted to validate the improvements of the proposed methods.


Weakness:
1. The motivation and the principle are not clear. Why do you use the prototype-learning predictor? What are the inherent differences between the linear predictor and the prototype-based predictor? Why does this work? Line 124-130 only demonstrates the differences on the practice. It is better to analyze or provide more reasons behind the regularization.
2. It is better to provide more details about the prototype initialization and updating, which are confused now. What does the learned prototype represent? How to perform clustering on them to find out internal sub-classes? What is the definition of the distance between the  pixel  and the prototypes?
3. Why are the improvements on the blender setting of the PASCAL VOC 2012 incremental?
4. It is better to perform the ablation studies about the CutMix. What is the performance about using other strong augmentations?

---

> ### Author Response · Authors · 2022-08-02
> **Clarify motivation, why our method works and more ablation studies**
>
> Thanks for your positive review and valuable feedback! Below, we address your points individually.
>
> > Weakness1.1: The motivation and the principle are not clear.
>
> Motivation has been presented in Line1-7 and Line28-31 of our paper. Specifically, we target the problem of large intra-class variation and challenges of propagating pseudo labels in semi-supervised semantic segmentation. Our key idea is to regularize the feature representation to improve the label propagation process.
>
> > Weakness1.2: Why do you use the prototype-learning predictor? What are the inherent differences between the linear predictor and the prototype-based predictor? Why does this work? Line 124-130 only demonstrates the differences on the practice. It is better to analyze or provide more reasons behind the regularization.
>
> ''Inherent difference and why it works'' have been presented in Line 34-40, Line 124-130 and Line 170-183 of our paper. Moreover, the linear classifier has learnable parameters and can adapt to imperfect features to fit pseudo-labels. The prototype-based classifier does not involve learnable parameters and thus to match similar predictions of the linear classifier, the prototype-based classifier calls for better feature representation (see the example introduced in Line 170-183). Thus our cross-predictor consistency loss encourages better feature learning for semi-supervised learning.
>
> > Weakness2.1: It is better to provide more details about the prototype initialization and updating, which are confused now.
>
> Sorry for the confusion. Line 188-193 of our paper gives the details for prototype initialization, which consists of the following steps.
> 1.  Train the semantic segmentation network on the given limited fully-labeled samples.
> 2.  Use the trained segmentation network to extract feature representations of these labeled samples (i.e. the feature representation before feed into the classifier of DeepLabv3+ and perform interpolation on the feature representation to match the input image size). We then sample a certain amount of pixels with their representations for each category.
> 3.  Perform k-means clustering (other clustering methods are also possible) on sampled pixel representations from each category. This step creates K sub-classes for each category. We use the feature average of samples in each subclass to obtain the initial prototypes of each category.
>
> Meanwhile, we dynamically udpate the prototypes during the model optimization and Line194-205 presents the details of how prototypes are updated.
>
>  > Weakness2.2: What does the learned prototype represent?
>
>  Intuitively, the learned prototypes represent the centers of subclasses in each category. Please see the Figure 2 \(c\) of our newly updated paper.
>
>  > Weakness2.3: How to perform clustering on them to find out internal sub-classes?
>
> The sub-classes are identified by performing K-means clustering on pixel representations of each category from the labeled images only. More details are given in Line 188-193 and our response to weakness 2.1 of Reviewer sZ7j.
>
>  > Weakness2.4: What is the definition of the distance between the pixel and the prototypes?
>
> We use the cosine similarity between the feature representation of pixels and the prototypes in our paper. Please find the description in Line150 of our paper.
>
> > Weakness3: Why are the improvements on the blender setting of the PASCAL VOC 2012 incremental?
>
> Blender setting of the PASCAL VOC 2012 is constructed with a finely annotated dataset and an augmented coarsely annotated dataset. Given the labeled set is noisy, the semi-supervised semantic segmentation algorithms may be mis-guided and the performance will be compromised.
>
> > Weakness4: It is better to perform the ablation studies about the CutMix. What is the performance about using other strong augmentations?
>
>
> | Classic VOC (1/16 setting)         | Cutout[1] | ClassMix[2]|
> | :---                                  | :----: | ---: |
> | U2PL                                  | 66.82 | 67.77 |
> | Ours w/o prototype-based classifier   | 66.86 | 66.93 |
> | Ours                                  | 69.24 | 69.36 |
>
> We perform ablation studies about using other strong data augmentation in our approach by incorporating another two popular data augmentations: Cutout[1] and ClassMix[2]. According to the results shown in the above table, our method can still outperform other comparing methods. It verifies the effectiveness of our proposed prototype-based consistency regularization and indicates that it is not just for CutMix but should apply to any strong data augmentations.
>
> [1] Terrance DeVries and Graham W Taylor. ''Improved regularization of convolutional neural networks with cutout.'' CoRR, abs/1708.04552, 2017.
>
> [2] Viktor Olsson, Wilhelm Tranheden, Juliano Pinto, Lennart Svensson. ''ClassMix: Segmentation-Based Data Augmentation for Semi-Supervised Learning'', WACV2021

---

> ### Author Response · Authors · 2022-08-09
> **Sincerely Look Forward to Your Feedback**
>
> Dear Reviewer sZ7j,
>
> Thanks again for your insightful suggestions and comments. As the deadline for reviewer-author discussion is approaching. We are glad to provide any additional clarifications that you may need.
>
> We have carefully studied your comments and added additional clarifications and experiments in our previous responses to address your concerns. We genuinely hope you could kindly check our response.
>
> We hope that our previous responses have convinced you the merits of our work. Please do not hesitate to contact us if there are other clarifications or experiments we can offer.
>
> Thank you for your time again.
>
> Best wishes,
>
> Authors

---

### Author Response · Authors · 2022-08-02
**General Response**

We thank the reviewers for providing valuable and thoughtful comments on our paper. Based on the reviews, our paper has been mainly revised from the following perspective views:

- **Prototype visualization.** We revise the t-SNE figure of our method through visualization prototypes in the same space of pixel representations.
- **Citation details.** We add the detail publication information of the references cited in our paper.
- **Algorithm tables.** We add two algorithm tables for the prototype initialization and global view of our approach respectively for better undertanding.
- **Quantitative metric for intra-/inter-class discrimination.** We borrow the principle of linear discriminant analysis(LDA) and calculate the intra-/inter-class variance of the feature representations for each comparing methods.
- **Ablation studies.** We add two ablation studies to further inspect the effectiveness of our approach, i.e., various strong data augmentations and the confidence threshold.

Due to the space constraint of the main paper, we add the first two revisions to the main paper and the latter three revisions to the supplementary material.

We hope that our responses have fully addressed all of the reviewers' concerns and remain committed to clarifying any further questions that may arise during the discussion period.

---

### Meta-Review · Area_Chair_zD61 · 2022-08-31

**Recommendation:** Accept
**Confidence:** Less certain

**Metareview:**

This paper proposes a teacher-student scheme for semi-supervised semantic segmentation. A consistency regularization is setup between a prototypical classifier and a linear classifier and different augmentation degrees (weak vs. strong) are applied to the teacher and student networks. On the positive side, the reviewers have found the ideas in this paper simple and strong in practice and they have indicated that the proposed setting is interesting. While the novelty of this paper may seem incremental since consistency regularization, in general, is heavily explored in semi-supervised training, the proposed setting is new for the semantic segmentation problem. One of the main criticisms of this submission is that it consists of many moving parts that are not well motivated and how they are orchestrated during training is missing from the original submission. After careful discussion, I believe that the merits of this submission outweigh the issues, and I am happy to recommend this paper for acceptance.

Last but not least, I strongly recommend the authors bring the algorithms to the main (if possible), provide additional implementation details, and make their code publicly available.

**Award:**

No

---

### Decision · Program_Chairs · 2022-09-14

Accept